# Differences in Habitual Physical Activity Behavior between Students from Different Vocational Education Tracks and the Association with Cognitive Performance

**DOI:** 10.3390/ijerph18063031

**Published:** 2021-03-16

**Authors:** Rianne H. J. Golsteijn, Hieronymus J. M. Gijselaers, Hans H. C. M. Savelberg, Amika S. Singh, Renate H. M. de Groot

**Affiliations:** 1Faculty of Educational Sciences, Open University of the Netherlands, 6419 AT Heerlen, The Netherlands; h.gijselaers@gmail.com; 2Department of Nutrition and Movement Sciences, School for Health Professions Education (SHE), School for Nutrition and Translational Research in Metabolism (NUTRIM), Maastricht University, 6229 GT Maastricht, The Netherlands; hans.savelberg@maastrichtuniversity.nl; 3Mulier Institute, 3584 AA Utrecht, The Netherlands; a.singh@mulierinstituut.nl; 4Center for Physically Active Learning, Faculty of Education, Arts and Sports, Western Norway University of Applied Sciences, 6851 Sogndal, Norway; 5Department of Complex Genetics, Faculty of Health, Medicine and Life Sciences, School for Nutrition, Toxicology and Metabolism, Maastricht University, 6229 ER Maastricht, The Netherlands

**Keywords:** adolescents, accelerometry, executive function, vocational education, physical activity, sedentary behavior

## Abstract

Vocational education and training (VET) educates students for a broad range of occupations, which may be associated with differences in habitual physical activity behavior (PAB). Research suggests that physical activity (PA) is positively and sedentary behavior (SB) is negatively associated with cognitive performance. Therefore, we aimed to compare habitual PAB in VET students from different educational tracks and investigate its association with cognitive performance in a cross-sectional study. Students wore an ActivPAL accelerometer continuously for seven days to measure PAB. Cognitive performance was assessed with objective tests for inhibition, shifting, and updating. *Hairdresser* and *Sports* students sat significantly less than *Administrative* and *Nursing* students. *Hairdresser* students stood significantly more than other tracks. *Admin* students stood significantly less than *Sports*/*Nursing* students. *Sports* students moved significantly more than *Hairdresser* and *Nursing* students. Time in bed was significantly lowest in *Nursing* students. No significant associations between any PAB and cognitive performance were found. In general, *Admin* students showed the unhealthiest habitual PAB. Higher PA or lower SB neither improve nor decrease cognitive performance. Thus, future health interventions focusing on exchanging SB for PA at schools can facilitate a healthier lifestyle of VET students, especially in Admin students, without interfering with cognitive performance.

## 1. Introduction

It is known that physical activity (PA) has numerous physical health benefits, both for adults [1] and adolescents [2,3]. Besides health outcomes, the benefits of PA for cognition and brain function, which may facilitate learning and academic performance [4,5,6,7], are increasingly studied [8]. In adults, independently of PA, sedentary behavior (SB) affects health negatively [9,10]. Furthermore, SB is also associated with lower cognitive performance [11]. To date, such associations are less extensively studied in adolescents [3,12,13,14]. In this study, the combination of PA and SB is defined as physical activity behavior (PAB). Little is known about PAB and cognitive performance in the context of vocational education and training (VET), as most studies focused on primary or secondary schools, or university students [8,15,16]. Therefore, research on PAB and its association with cognitive performance specifically in the VET setting is warranted. 

Regular participation in PA is associated with better physical, mental and cognitive health in adolescents [17]. Moreover, independently of PA, high amounts of SB are associated with decreased physical health, self-esteem, academic achievement, and unfavorable pro-social behavior [12,13]. Thus, adopting healthy PAB is important for VET students, and insights into actual PAB can provide guidance for future interventions. Yet, research regarding PAB specifically in VET students is scarce, despite representing a substantial part of the student population, as 40% of all Dutch students are enrolled in VET [18]. It is important to consider that there is a lot of variation in curricula and duration of VET programs (e.g., from hairdressing to the army, from information and communication technology (ICT) to nursing). Such programs educate students for specific vocational jobs and direct introduction into the labor market [15]. Hence, it is likely to expect distinct differences in the PAB of students from different educational tracks. For example, hairdresser track students probably stand more and sports track students probably move more as, for example, compared to administration track students, who probably sit a lot during the day. In addition, students spend a significant amount of time at internships, where they are trained on their future job. Subsequently, it is warranted to examine the presence of differences between educational tracks when studying PAB in the VET setting.

Although only few studies distinguished between educational tracks, percentages of VET students that are sufficiently physically active are low and range from 12 to 49%, with boys being significantly more physically active than girls [15,19,20,21,22,23]. However, it should be noted that thresholds for sufficient PA varied from 150 min of PA per week to 60 min of PA per day, based on PA guidelines for youths or adults. Carl et al. [24] compared nursing students and automotive mechatronics students, and reported the latter to be less physically active, whereas Haug et al. [15] ranked vocational fields regarding the prevalence of physical inactivity and found lower levels of inactivity among construction students and higher levels in beauty students. Several studies report that PA in VET students is lower compared to students in other school types (e.g., secondary school, higher education) [20,21,25].

Studies reporting on SB in VET students are even more scarce. One study reported general sedentary time (i.e., time spent lying or sitting down) to be almost 9 h of waking time and found IT students to be more sedentary than nursing, business and administration, and hotel, restaurant and catering students [23]. Older studies assessed screen time as a proxy for SB and indicated that 51 to 80% of VET students did not meet the guideline of less than two hours of screen time per day (e.g., television viewing, playing video or computer games) [20,26]. However, due to the increasing use of laptops, tablets and smartphones (either study-related or for leisure), such measures and guidelines may be outdated, as VET students probably spend even more time using screens and thus being sedentary. Additional insights into overall SB are warranted, especially since most of the time at school is spent sitting and thereby has a major contribution to total sedentary time [27]. More specific insights into both PA and SB conjointly can guide the focus of specific PAB interventions for specific educational tracks. 

Since PAB is also associated with cognitive performance in children and older adults [8], we were also interested in this association in VET students, as cognitive performance is associated with academic achievement, career success and mental health [28]. One of the domains frequently studied in relation to cognitive performance is executive functioning (EF) [29], which incorporates cognitive processes such as inhibitory control, working memory and cognitive flexibility [30]. These processes take place in the prefrontal cortex, which is known to mature into early adulthood. Consequently, insights into the associations between EF and PAB in (adolescent and young adult) VET students are valuable [16]. For children and pre-adolescents, positive effects of PA on EF, attention, math and academic performance are reported [7,8]. Additionally, a review that considered both children and adolescents found a small but significant effect size of PA interventions on EF [31]. Although reviews specifically conducted in adolescents and young adults were limited and showed mixed results, evidence was regarded promising but insufficient to draw conclusions on the association between PA and EF [8,16,29]. According to our knowledge, there are no studies examining the association between PA and EF in VET students specifically, thus requiring further research. 

In general, the association between SB and cognitive performance is studied less extensively. According to a systematic review in adults, SB was associated with lower cognitive performance [11]. However, a systematic review in children and adolescents found inconclusive evidence for the association between objectively measured SB and cognitive outcomes [14]. A recent study that examined combined associations of SB and PA in children showed that low SB and high PA were positively associated with EF. Moreover, even for those that are physically active, a high amount of SB is negatively associated with EF [32]. Nonetheless, research regarding SB and EF in adolescents and young adults, and VET students specifically, is scarce and warrants further research. 

Thus, this study aimed to investigate differences in habitual PAB in VET students in various educational tracks. Moreover, since an association has been suggested between PAB and cognitive performance [4,5,6], we also investigated this association in our study sample. Since most of the above-mentioned studies relied on self-reported measures of PA, which are prone to over-reporting, we used an accelerometer to acquire a more objective insight into PAB. It was expected that clear differences in habitual PAB were present between various educational VET tracks. In addition, it was expected that PA was positively and SB was negatively associated with cognitive performance. 

## 2. Materials and Methods

### 2.1. Design

This study is part of the PHIT2LEARN study—PHysical activity InTerventions to enhance LEARNing—a research project in which PA and SB are studied in relation to EF and academic performance in vocational education and training (VET) students. The current study had a cross-sectional observational design and was executed at an institute in the central south of the Netherlands that offers the full range of VET education. The PHIT2LEARN study is funded by the National Regieorgaan Onderwijs (Netherlands Initiative for Education Research) by grant number 405-16-412. Data from this part of the PHIT2LEARN study are available and stored permanently on DANS EASY, a sustainable platform for archiving research data [33]. This study is registered in the Dutch Trial Register (NL6573). 

### 2.2. Participants

Participants were recruited from 19 classes of a VET institute in the central south of the Netherlands. These classes represented different study years, different educational subjects, and different VET levels (i.e., in VET, educational programs are offered at four-degree levels: 1, assistant level; 2, basic vocational level; 3, full professional level; 4, specialist level). Students were divided into four tracks: Hairdresser, Admin (i.e., secretary or ICT), Nursing (i.e., healthcare setting), and Sports (i.e., sports and movement instructor), conforming to the corporate categorization applied by the VET institute. Participants could be included if they were studying in one of the classes that were invited to participate in this study. There were no specific exclusion criteria as we wanted to include a general population. Factors that could potentially influence the outcomes of the study were measured and taken into account in the analyses, as discussed below. 

A sample size calculation was performed in G*Power (Universität Düsseldorf, Düsseldorf, Germany) and a sample size of 76 participants was needed for a power of 0.8 and assuming a large effect size of 0.40. A large effect was expected regarding the differences in habitual PAB between the various tracks due to the nature of the profession they are educated for.

### 2.3. Procedure

The study was approved by the Ethical Research Board of the Open University in the Netherlands (U2017/00519/FRO). The management of the VET institute approved the execution of the study at their school. All potential participants received verbal and written information about the study during an information session. Subsequently, students were given at least one week to consider their participation in the study. They also received an informed consent form which they signed and handed in before the start of the first session when they wanted to participate in the study. During two sessions, at least 7 days apart, all data were collected.

During session one, all participants were asked to attach an accelerometer to their leg to measure PAB. Further, height and weight were measured. All participants completed a questionnaire regarding general information (e.g., sex and age), education and the education of their parents, health, pubertal status, and lifestyle (i.e., sports, way of commuting, smoking, food, alcohol and drug intake). During session two, at least seven days after session one, all students participated in two tests measuring their cognitive performance. Each test was preceded by a test explanation for the entire class. Each new part of the cognitive tests consisted of a practice round and a test round. Two researchers continuously monitored the situation in the class, and issues that could influence the test results were written down. This information could clarify any odd results. After completing both tests, the students handed in their accelerometer. 

### 2.4. Materials

#### 2.4.1. Physical Activity Behavior

To measure PAB during a normal school/internship week, all participants taped a waterproofed ActivPAL3^TM^ (PAL Technologies Ltd., Glasgow, Scotland, UK) accelerometer on the midpoint of the anterior part of their right thigh, using Tegaderm^TM^ (3M, St. Paul, MN, USA) transparent film roll. The ActivPAL3^TM^ is a small device (53 × 35 × 7 mm) that measures bodily accelerations and identifies the wearer’s posture. Making the accelerometers waterproof allows continuous measurement during the week, without breaks for showering, swimming, or sauna, and ensures adherence of wearing the accelerometer. Data were recorded at 20 Hz and summarized in 10-s time intervals (epochs). The validity and reliability of the ActivPAL3 for usage in free living situations were shown previously, in adults as well as younger people [34,35]. This was shown by an overall agreement between observer and ActivPAL3 of 95.9 percent, and an inter-device reliability ranging from 0.79 to 0.99. 

Participants were asked to wear the accelerometer continuously for at least seven days. The period over which the accelerometer was worn differed between study disciplines as some educational tracks had alternating school and internship weeks. To incorporate both school and internship weeks and obtain accurate representations of PAB, these students wore the accelerometer for 10 or 11 consecutive days. All other students, who had rather constant, similar weeks during the school year, wore the accelerometer for seven consecutive days. 

Data were processed with ProcessingPAL [36] and considered usable and valid when at least two weekdays and two weekend days were recorded, as it was shown that this is the minimum requirement for valid PAB parameters [37]. From all valid days, the average per day per outcome was calculated. Outcome measures included time in bed and time spent sitting/lying, standing, and stepping (i.e., this included all movement). 

In addition to PA and SB measures, the SB-ratio was also calculated. The SB-ratio represents the ratio between time spent sedentary in bouts shorter than 30 min and total time spent sedentary. Thus, the SB-ratio is an indicator of interrupted sitting behavior; the higher the ratio, the more time spent sedentary was spent in bouts of maximum 30 min. The 30 min cut-off is based on the recommendation to interrupt SB every 30 min [38]. 

In addition, adherence to the Dutch national PA guideline [39] was calculated, using moderate-to-vigorous (MVPA) stepping time derived from the ActivPAL, based on a step rate of ≥100 steps/minute corresponding with ≥3 METs (metabolic equivalents) [40,41]. The Dutch PA guideline is directed at either youths (until the age of 17; minimum 1 h of MVPA per day, calculated for every uniquely measured day) or adults (18 and older; minimum 150 min of MVPA per week, calculated as average per week, based on all valid data). The majority of VET students in the Netherlands are aged between 16 and 22 [42], and thus vary around the age cut-off applied for the Dutch PA guideline. Therefore, adherence to guidelines for both youths and adults was calculated. 

#### 2.4.2. Cognitive Functioning

Cognitive functioning was measured by means of updating, inhibition, and shifting, using Miyake and Friedman’s framework [43,44,45]. To cover these three aspects, the letter-memory test (LMT) and color-shape test (CST) were used. Using Inquisit Web version 5.0.7 (Millisecond Software, Seattle, WA, USA), both tests were conducted digitally via the students’ own laptop or a laptop provided by the researchers in case the student did not have a (functioning) laptop. The tests were conducted simultaneously for all students from a specific class in a face-to-face classroom setting. The standard scripts of both tests were used from the Millisecond Test Library. Using Inquisit Lab, instructions for the tests were translated and tailored to the study population. 

The LMT [45] assesses the updating capacity of the working memory. Participants completed three training trials, followed by 12 test trials. In each trial, the participant was shown a series of five, seven or nine consonants, one letter at a time. All letters were shown in the middle of a laptop screen for 2500 ms. Each length of the letter series was used just as often. At the end of each series of letters, the students’ task was to recall the last three presented letters by clicking these in a provided letter matrix. As the participants did not know the length of each letter series, they constantly had to recall the last three letters. The entire task had to be completed as accurately and quickly as possible. In case participants did not remember the last three letters, they could click “Blank” for the letters they did not recall. In some cases, participants continued without entering the last three letters or “Blank”, which left some missing values. Figure 1 shows a visual representation of the LMT. Accuracy of LMT performance was measured by the total number of correctly recalled letters, regardless of the order, which is a measure for the updating capacity of the working memory. 

The CST [44,45] consists of five different rounds, enabling the possibility to test inhibition capacity as well as shifting capacity. Each round consists of 16 training trials and subsequently 64 test trials. In each round, participants were presented shapes (a triangle or a circle), colors (red or green) or combinations of both (a triangle on a red background, a triangle on a green background, a circle on a red background, or a circle on a green background). All stimuli were presented in the middle of a laptop screen until the answer was entered. In each round, participants were instructed to answer with the color or the shape of a presented stimulus. Answering occurred via the A-key for circle and red, and the L-key for triangle and green, using the index fingers. Each task was to be completed as accurately and quickly as possible. Figure 2 shows a visual representation of the CST.

In round one (shape-test), participants were presented with shapes and answered whether they saw a triangle or a circle. In round two (color-test), participants were presented with colors and answered whether they saw a red or a green color patch. Round three (shifting-test) was a combination of both previous rounds; participants were alternately presented with a shape or a color. The presentation of shapes and colors was random, which means that in some cases a task was repeated (i.e., two color-trials after each other), and in other cases the participants needed to switch between the shape-task and the color-task (i.e., shifting; a color-trial after a shape-trial, or the other way around). Depending on the stimulus, the participants needed to answer with the shape or color they saw. In round four (shape-inhibition-task) and five (color-inhibition-task), shapes superimposed on color patches were presented. The respective tasks were to answer with the shape (round four) and color (round five), regardless of the other distracting stimulus. For all rounds of the CST, reaction times (RTs) and correctness of the answers were recorded. RTs were based on correct trials only. 

Data were first filtered on correctness by chance, which means that participants who had rounds with 32 or less correct answers (out of 64) were excluded from data analyses. To yield only real shifting performance in the shifting-test, trials that did not appeal for shifting between the shape- and the color-task (i.e., two consecutive shape or color trials) were excluded from the calculation for the outcome variable. Next, all congruent trials in the inhibition-tasks were excluded from the data. Congruent trials are test trials that have stimuli that are in agreement with each other (i.e., a triangle on a green color patch or a circle on a red color patch) and require the same response, hence they do not require inhibition. Trials with RTs lower than 170 milliseconds or higher than 5000 milliseconds were excluded from analyses [44]. The lower limit of 170 milliseconds is based on the lowest RT on a non-cognitive task (i.e., simple RT) [46]. RTs lower than 170 milliseconds on a cognitive task indicate that the participant did not absorb the stimulus and probably just pressed the response key, by accident or due to a lack of interest in the test. Finally, data were filtered on intrapersonal outliers with the mean plus or minus three SD. Since the mean RT of participants could be substantially distorted by only one trial with a very high RT, intrapersonal filtering was applied. Excluding only this trial, instead of all data of this participant, could already yield valid data. The data processing described here is based on the original article published on this test [44]. Only the lower limit was altered from 100 to 170 milliseconds and that threshold is still very conservative. The outcome measures of the CST are inhibition cost and shifting cost. Shifting cost was calculated by subtracting the mean RT in round one and two from the mean RT in round three, which is the shifting-task. Inhibition-cost was calculated by subtracting the mean RT in round one and two from the mean RT in round four and five, which are the inhibition tasks.

#### 2.4.3. Questionnaires and Other Measures

During both sessions, the students filled out a questionnaire regarding background variables that could influence their cognitive performance and/or physical activity behavior. Health and medical issues, including medication use, were measured via a short questionnaire.

Medication use was categorized as not reported, directly impairing cognitive performance, indirectly impairing cognitive performance by potentially causing sleepiness, dizziness, or a headache, or, unknown to affect cognitive performance. Learning disabilities were asked in a similar manner and were categorized as not reported, or, potentially related to cognitive performance. Disabilities that affected PAB were also asked in a similar manner and were categorized as either reported or not reported. Participants also indicated whether they had color-blindness. Height was measured in meters. Weight, visceral fat, fat, and muscle percentages were measured without shoes, with empty pockets and heavy vests or sweaters removed with the KaradaScan, a body composition monitor (Omron, BF511). Height and weight were used to calculate body mass index (BMI; i.e., weight in kilograms divided by height in meters squared). Visceral fat (fat around organs) is measured on a scale ranging from 1–30, where 1–9 is normal, 10–14 is high, and 15–30 is very high. 

### 2.5. Statistical Analyses

Analyses were performed with SPSS (version 24.0; SPSS, Inc., Chicago, IL, USA). The level of significance was set at 0.05 and 0.1 for borderline significance. After data inspection and data cleaning, descriptive statistics were provided and compared between groups, using One-way Analyses of Variance (ANOVA) and chi-square tests. Differences between educational tracks in all habitual PAB measures and in SB-ratio were assessed with One-way Analyses of Covariance (ANCOVA), controlling for potential covariates. Effect sizes (i.e., Cohen’s *d*) with 95% confidence intervals (95% CI) were calculated, with effect sizes of 0.20, 0.50, and 0.80 indicating small, medium and large effects, respectively [47]. Chi-square tests were run to investigate differences in guideline adherence between educational tracks. A Bonferroni correction was applied to all post-hoc tests. Three linear regression analyses were conducted to assess whether an association was present between the PAB variables and either inhibition, shifting and updating. Zero-order correlations were run to evaluate the association between habitual PAB measures, SB-ratio, guideline adherence and the cognitive outcomes without controlling for other variables. 

## 3. Results

### 3.1. Participant Characteristics

In total, 374 VET students were invited to participate in the study. Of these, 230 (62%) agreed to participate. Valid PAB data were available for 103 students. Further screening led to exclusions on the shifting measure (n = 3; very high shifting values, more than 3 SD from the mean) and the updating measure (*n* = 1; very low value, more than 3 SD from the mean). Further inspection revealed that 10 participants had no data on any of the three outcome variables, and this was due to the filtering of the cognitive data according to all filtering steps (see section *Cognitive Performance* under *Materials and Methods*). The data set used to investigate the association between PAB variables and cognitive outcomes thus included 93 students. For the PAB analyses, the data set with 103 students was used. Where cognitive data were missing, this is reported with the number of cases in each specific analysis below. 

Of the 103 students, 71 (68.9%) were female and the mean age was 19.2 ± 3.2 years. Table 1 shows the distribution of students over the four educational tracks and the differences in demographics and body composition variables. Students in the various tracks differed with regard to age, sex, fat percentage and muscle mass percentage, as shown in Table 1. With regard to health and medical issues, one quarter of all students (*n* = 26, 25%) reported medicine use that might influence cognitive performance; 23 (22%) students reported a learning disorder that might influence cognitive performance; 9 (9%) students reported a disability that influenced their physical activity; and four (4%) students reported color-blindness, which might influence performance on the CST. These medical issues were not significantly different between the various educational tracks.

### 3.2. PAB in VET and Difference between Educational Tracks

Differences in PAB were controlled for age, gender and fat percentage, but not for muscle mass percentage, since the latter two are highly correlated and thus may cause multicollinearity. Muscle mass percentage was omitted based on a high variance inflation factor. All means reported below are estimated marginal means controlled for age, sex, and fat percentage.

All activity types significantly differed between educational tracks: sedentary time, *F*(3, 92) = 13.45, *p* < 0.001; standing time, *F*(3, 92) = 12.17, *p* < 0.001; stepping time, *F*(3, 92) = 11.18, *p* < 0.001; and time in bed, *F*(3, 92) = 5.86, *p* = 0.001. Bonferroni corrected pair-wise comparisons indicated significant differences in PAB between the educational tracks (Figure 3). All differences showed large effect sizes (Table 2).

#### 3.2.1. Sedentary Time

Both Hairdresser (M = 7.65, SE = 0.29) and Sports track students (M = 7.86, SE = 0.26) sat significantly less hours per day than Admin (M = 9.76, SE = 0.29) and Nursing track students (M = 9.23, SE = 0.36). Sedentary time was highest in Admin track students.

#### 3.2.2. Standing Time

Standing time was highest in Hairdresser track students (M = 5.14, SE = 0.23). They stood significantly more hours per day than students in Admin (M = 3.12, SE = 0.23) and Sports (M = 3.94, SE = 0.21) tracks. The difference with Nursing track students was borderline significant (M = 4.23, SE = 0.29). In addition, Nursing track students also stood significantly more hours per day than Admin track students. The difference between Sports and Admin track students was also borderline significant.

#### 3.2.3. Stepping Time (All Movement)

Movement was highest in Sports track students (M = 2.61, SE = 0.12). They moved significantly more hours per day than students in Hairdresser (M = 1.72, SE = 0.13) and Admin (M = 1.78, SE = 0.13) tracks. Differences with Nursing track students (M = 2.16, SE = 0.16) were not significant. 

#### 3.2.4. Time in Bed

Time in bed was lowest in Nursing track students (M = 8.37, SE = 0.24). They spent significantly less time in bed than students in all other tracks (Hairdresser: M = 9.50, SE = 0.19; Admin: M = 9.34, SE = 0.19; Sports: M = 9.59, SE = 0.18). 

#### 3.2.5. SB-Ratio

The SB-ratio differed significantly between the four educational tracks, *F*(3, 92) = 6.22, *p* = 0.001. Bonferroni corrected pair-wise comparisons revealed that Sports track students (M = 0.587, SE = 0.020) have a significantly higher SB-ratio compared to Admin track students (M = 0.468, SE = 0.022), with a large effect size (Cohen’s *d* = 1.06 (95% CI = 0.51–1.61, *p* < 0.001)). The SB-ratio of Sports track students is also significantly higher than the SB-ratio of Hairdresser track students (M = 0.496, SE = 0.022), also with a large effect size (Cohen’s *d* = 0.81 (95% CI = 0.28–1.35, *p* = 0.03)) (Figure 4). This means that Sports track students, compared to Admin and Hairdresser track students, spent a significantly larger proportion of their sedentary time in bouts of maximum 30 min. Thus, Sports track students have significantly shorter bouts of SB than Admin and Hairdresser track students. Differences with Nursing track students (M = 0.494, SE = 0.027) were not significant.

#### 3.2.6. Adherence to PA Guidelines

Of all 103 students in the sample, irrespective of age, 15% (*n* = 15) adhered to the PA guideline for youths (i.e., minimum 1 h of MVPA per day). In contrast, all students (100%) adhered to the guideline for adults (i.e., minimal 150 min of MVPA per week). From those under 18 (*n* = 40), only 7 (18%) adhered to the guideline for youths. When looking at differences between study tracks (Figure 5), chi-square tests in which the PA guideline for youths was applied to all students indicated that guideline adherence differed significantly between tracks, *χ*^2^ (3) = 9.35, *p* = 0.03. The percentage of Sports track students meeting youth PA guidelines (29%) was significantly (*p* = 0.003; Bonferroni corrected) different from all other tracks (Hairdresser: 8%; Admin: 4%; Nursing: 11%). The other tracks did not differ significantly from each other. Since all students adhered to the guideline for adults, there were obviously no differences between study tracks.

### 3.3. PAB and Cognitive Performance

Zero-order correlations between any of the cognitive variables (i.e., parametric and non-parametric) were low and non-significant, thus each of the cognitive tests measured a different aspect of EF. Regression analyses showed that none of the PAB variables (i.e., sitting, standing, stepping) was significantly associated with any of the cognitive variables. Additionally, zero-order correlations also showed no significant correlations between any of the PAB variables and any of the cognitive variables (Table 3). 

Additionally, it was checked whether differences regarding students’ EF were present between the educational tracks. ANOVAs for all three cognitive variables were not significant (all *p* > 0.29). Zero-order correlations (i.e., verified parametric and non-parametric) showed no significant association of the SB-ratio with any of the cognitive outcome variables (all *p* > 0.81, parametric), thus the interruption of prolonged sitting was not associated with EF. Zero-order correlations (i.e., point-biserial) showed no significant association of guideline adherence with any of the cognitive outcome variables (all *p* > 0.84).

### 3.4. Covariate Analyses

Subset analyses showed that the associations between PAB and EF were not different when excluding participants by medication use, learning restriction, physical activity restriction, color blindness, and age; all findings were similar to the results reported above. 

## 4. Discussion

The aim of this study was twofold: to investigate differences in habitual PAB in VET students in various educational tracks and to evaluate whether PAB was associated with cognitive performance. 

### 4.1. Differences in PA between VET Students from Various Educational Tracks

Admin track students sat down the most, while Hairdresser students stood the most, and movement was highest in Sports track students. Sports track students showed more interrupted SB compared to Admin and Hairdresser track students, as indicated by the SB-ratio. In addition, the fact that Sports track students were more physically active than most other tracks is also reflected in the fact that a significantly higher percentage of students adhered to the PA guideline for youths. These clear and distinct differences can possibly be explained by differences in curricula or by the study choice of students being related to their PAB. Sports track students that need to acquire proficiency in different sports will probably have more lessons in which they practice sports than, for example, Admin students who specialize in ICT. Moreover, during practical lessons, Hairdresser students will stand most of the time. Furthermore, Nursing track students may be more aware of a healthy lifestyle and healthy PAB because this is addressed as part of their education [48,49]. Thus, the differences that we found were typical for the nature of the educational tracks, as a large part of the curriculum in VET entails the practice of professions VET students are educated for [50]. Thus, the educational setting may be an important factor in the habitual PAB of students [51]. On the other hand, the choice of a certain study track could also be related to the PAB behavior of students [52]. We could imagine that sports enthusiasts might be more likely to choose an education in sports. Consequently, Sports track students may be more involved in sports and exercise in their leisure time and thus show higher amounts of MVPA. Thus, besides differences attributable to curricula, PAB patterns could also reflect the preferences of students, independent of curricular PAB. 

Yet, since PAB reflects the nature of the occupation students were trained for, this could imply that typical educational track-related behavior is not compensated for during leisure time to level out differences between tracks: e.g., Sports track students are not more sedentary during leisure time to compensate their activity during school hours and Admin students are not more physically active during leisure time to compensate the high amount of sedentary time during school hours. This is in line with the findings regarding occupational and leisure-time PA in working populations, which report no compensatory effects of high occupational PA or SB during leisure time [53,54,55]. Thus, it seems important to address PAB during school hours. 

To the best of our knowledge, only one study objectively assessed PAB patterns in different educational VET tracks [23]. That study included educational tracks that partly differed from ours, but also reported that Nursing track students were more physically active and less sedentary than ICT students. Additionally, a meta-analysis of Prince et al. [56] showed differences in PA and SB across occupational groups similar to PAB differences between educational tracks in our study. Current VET students will represent a major part of our future working force, vulnerable to health risks and work-related diseases that are preventable by sufficient PA [24,50]. PA patterns are developed and consolidated during the transition from adolescence to young adulthood, and thus also during the transition from school to working life [2,50,57]. Accordingly, if healthy PAB can already be established during education for vocational jobs, this may have a positive influence on health during further maturation/ageing. Therefore, our results pinpoint future interventions that might have the highest potential to yield benefit.

Adherence to PA guidelines for youths was highest in Sports track students (29%), yet generally low, whereas all students adhered to the guideline for adults (i.e., applicable from 18 years old onwards). Based on low adherence to the PA guideline for youths, we could argue that in our study PA levels of VET students are low, in line with previous studies [15,19,20,21]. However, when applying adult guidelines, our total study population would be sufficiently physically active. Yet, because VET students are so close to 18 and current PA guidelines only consider PA, it may be more interesting to look at 24-h guidelines that take into account PAB from a broader perspective. Such guidelines have previously been established for youths; Canada recently also released a 24-h guideline for adults. This guideline recommends limiting sedentary time to 8 h a day and to break up sitting time as often as possible, to be physically active for 150 min per week (and 60 min per day for youths) and to spend several hours in light PA (including standing; LPA) [58]. In light of such a 24-h guideline, it can be noted that although the population of the current study is moving and standing throughout the day, the amount of sedentary time is still quite high (i.e., 7.53 to 9.80 h/day). Therefore, interventions should be aimed at replacing sedentary time by time spent in LPA or MVPA. 

Accordingly, interventions directed at reducing sedentary behavior in the classroom, as time in school is mostly spent sedentary, might be a promising avenue [27,59,60]. In addition, there are substantial differences in SB. Of all students, the Admin track students displayed most SB, least breaking up of SB, and least standing and PA. In line with PA guidelines, this group could potentially benefit the most, but may also be the most difficult to motivate to change their PAB. Therefore, interventions grounded in motivational theories may be required to establish behavior change.

### 4.2. PAB and Cognitive Performance

We hypothesized that PA was positively and SB was negatively associated with cognitive performance. However, no associations between any of the PAB variables and cognitive performance were found. 

Although the literature suggests relations between PAB variables and EF, there may be several explanations for not finding any associations between PAB and EF in the current study. As mentioned, most evidence for associations between PA and EF originates from children and older adults facing either cognitive development or decline, respectively [8,29]. As the pre-frontal cortex, which is responsible for EF, is known to be the last part of the brain to mature during adolescence and young adulthood, cognitive health is suggested to peak during young adulthood [61,62]. As a result, EF may reach an optimal level in which there is little room for further improvement, and thus it may be difficult to detect any differences [62]. Furthermore, in their systematic review, Cox et al. [29] noted that studies that found an association between PA and EF concerned more vigorous PA or showed differences in physical fitness, i.e., not duration. Therefore, intensity may be the key factor to elicit a measurable change in young and healthy participants. As higher intensity PA might also provide a physical fitness benefit, this may also be an important factor. In addition, most meta-analyses and systematic reviews that found an association between PA and EF also considered MVPA [8]. However, a recent systematic review found a trend toward LPA and learning, but high-quality evidence is currently lacking [63]. In the current study, we did not make a distinction between LPA and MVPA and considered all movement as PA. For future studies, it may therefore be important to make a distinction in PA intensity and to take physical fitness into account.

In the current study, we did not find any associations between SB and EF. As mentioned in the introduction, studies regarding such associations are limited. In addition, most of these studies used self-reported measures of SB, and the vast majority of studies that reported an association used screen time as a proxy for SB. By using an objective measure of SB, as we did in the current study, it is not possible to make a distinction between cognitively engaging SB (e.g., time spent doing homework or reading) or SB that does not require any mental effort (e.g., TV viewing) [64,65,66,67]. This is in line with a study of Syväoja et al. [68], who found an inverse association between self-reported screen time and academic achievement, but did not find any association between objectively measured SB and academic achievement in the same population. 

The absence of an association between PAB variables and cognitive variables does not necessarily mean that PAB does not exert influence on the cognitive performance of students. Additional research is necessary to study if this is related to the operationalization and measurement of PA and SB (e.g., PA intensity, screen time vs. doing homework). Furthermore, experimental studies are required to examine whether there is an association between changes in PA and cognitive performance.

### 4.3. Strengths and Limitations

Since PAB in the VET population has not yet been studied extensively, our study contributes to providing insights into PAB in this specific population. Such information can guide future intervention development. By also making a difference between several educational tracks, we were able to show that PAB differs between tracks and is related to the occupation students are trained for, and thus which educational tracks are more at risk for future health problems. The fact that we used objective measurements of PAB is another strong point of this study, since the little evidence that is available is predominantly based on self-reported measures of PA.

However, employing the objective measurement of PAB also has the drawback that device-based measures do not provide insights into the context of SB, which can be regarded as a limitation. In addition, we were not able to obtain objective PAB measurements for about half of the students because students took off the ActivPAL due to different reasons. Therefore, for future studies we suggest using a combination of device-based measures for the time spent in SB and self-reported measures to describe the context of SB [67,68], and to examine options to increase compliance to ActivPAL wear protocols [69,70]. Additionally, cognitive performance tests may be prone to environmental influences (e.g., caffeine or food consumption, sleep quality) and we may not have been able to control such environmental differences. Yet, we have no reasons to expect differences between the groups for such factors. Furthermore, we applied a cross-sectional study design, which by no means provides any clarity on causality. Therefore, future experimental studies could provide insights into the associations between changes in PAB and cognitive performance, and thus whether PAB interventions are promising to enhance learning and academic performance.

## 5. Conclusions

In general, time spent sedentary is high among VET students, who represent a substantial and understudied part of the Dutch student population. Moreover, there are substantial differences in PAB between students from various VET tracks, with Admin students exhibiting the unhealthiest habitual PAB. An important conclusion from this study is that the educational track students follow is associated with PAB. Thus, the school setting might be important in promoting healthy PAB in VET students, and our results provide directions for selecting the desired target population. Furthermore, although higher PA or lower SB does not enhance cognitive performance, they also do not seem to decrease cognitive performance and thus ultimately learning performance at school. Therefore, we recommend that future health interventions for VET students should focus on exchanging SB for PA, for example, by interrupting sitting, to foster VET students’ future health.

## Figures and Tables

**Figure 1 ijerph-18-03031-f001:**
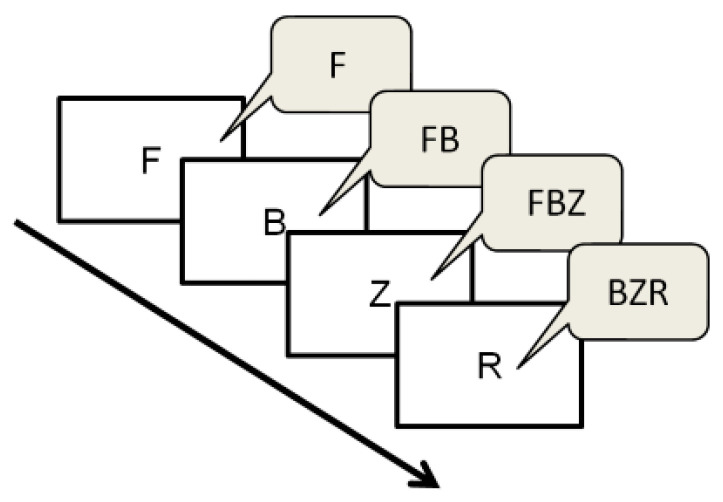
Visual representation of the letter-memory test. Each trial consists of a series of five, seven or nine consonants, one letter at a time, as depicted in the white rectangles. The students’ task is to constantly recall the last three letters shown, as depicted in the text balloons. At the end of the series, the student enters the last three letters he/she could recall.

**Figure 2 ijerph-18-03031-f002:**
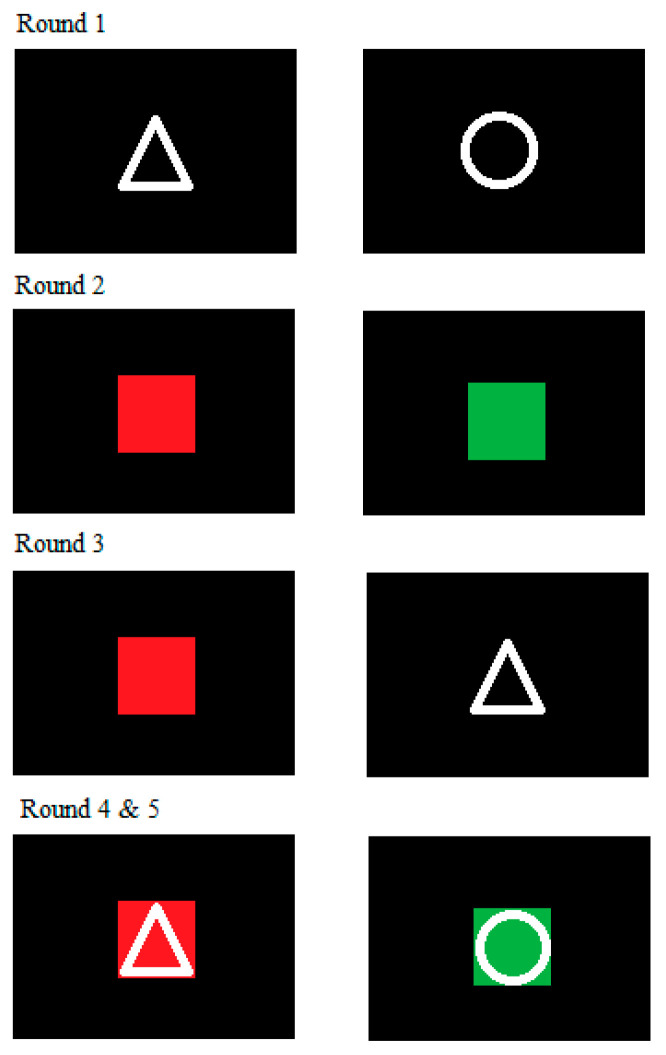
Visual representation of the color-shape test.

**Figure 3 ijerph-18-03031-f003:**
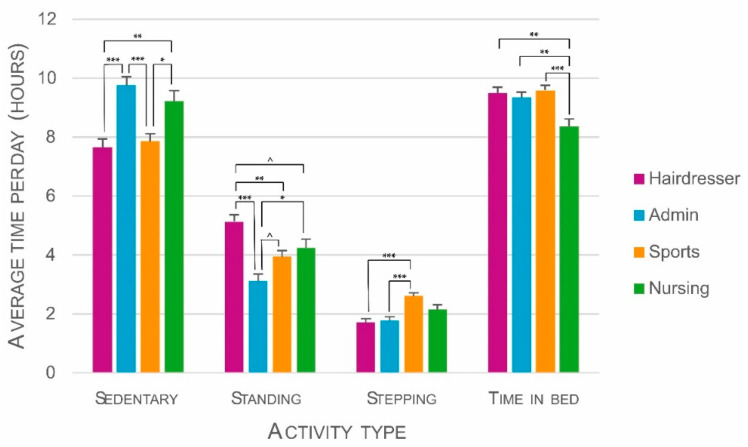
Differences between educational tracks in physical activity behavior (PAB) controlled for sex, age, and fat percentage. Error bars are standard errors. Significant Bonferroni corrected pair-wise comparisons are indicated with asterisks (*^ = p* < 0.1; * = *p* < 0.05; ** = *p* < 0.01; *** = *p* < 0.001).

**Figure 4 ijerph-18-03031-f004:**
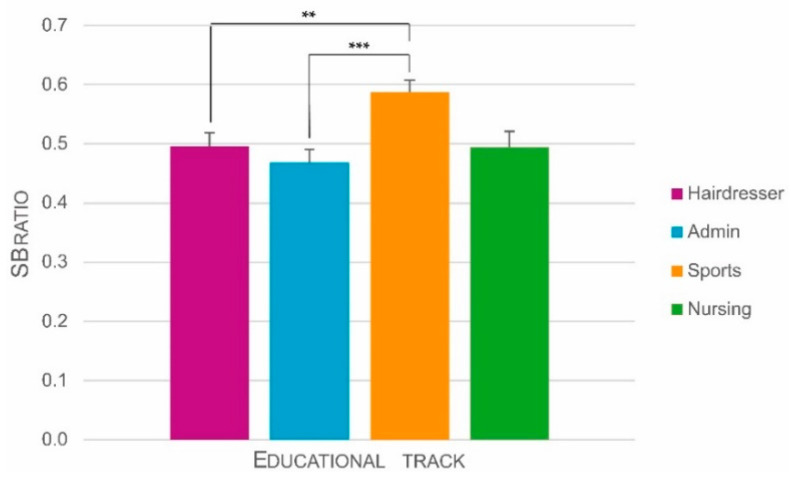
Differences between educational tracks in sedentary behavior (SB)-ratio controlled for age, sex, and fat percentage. Error bars are standard errors. Significant Bonferroni corrected pair-wise comparisons are indicated with asterisks (** = *p* < 0.01; *** = *p* < 0.001).

**Figure 5 ijerph-18-03031-f005:**
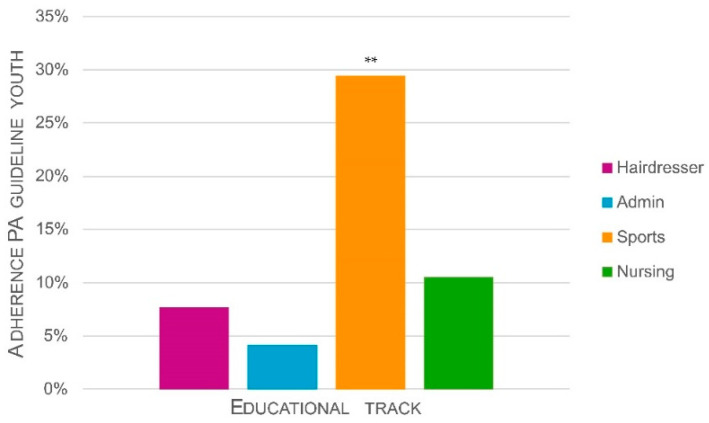
Differences between educational tracks in adherence to physical activity (PA) guidelines for youths. Significant Bonferroni corrected post-hoc tests are indicated with asterisks (** = *p* < 0.01).

**Table 1 ijerph-18-03031-t001:** Descriptive statistics and differences between educational tracks for demographics and body composition variables.

Characteristics ^1^	Total Group (*n* = 103)	Hairdresser (*n* = 26)	Admin (*n* = 24)	Sports (*n* = 34)	Nursing (*n* = 19)	*p*-Value	Post-Hoc Comparison ^2^
Mean age (SD)	19.2 (3.2)	19.5 (2.3)	18.6 (2.5)	17.6 (0.6)	22.6 (4.9)	<0.001	H, A, S < N; H > S
Female gender (%)	71 (68.9%)	26 (100%)	16 (66.7%)	13 (38.2%)	16 (84.2%)	<0.001	H > A, N; S < A, N
Mean BMI (SD)	22.9 (3.5)	21.8 (3.7)	23.8 (4.0)	22.7 (2.3)	23.4 (4.0)	0.22	-
Mean fat percentage (SD)	27.6 (9.1)	30.9 (7.2)	29.0 (9.9)	22.5 (7.3)	30.7 (9.7)	<0.001	H, A, N > S
Mean muscle mass percentage (SD)	32.0 (6.4)	28.3 (2.4)	31.1 (7.0)	36.4 (6.0)	30.1 (6.0)	<0.001	H, A, N < S
Mean visceral fat score (SD)	4.0 (1.9)	3.4 (1.2)	4.6 (1.7)	4.9 (1.4)	4.1 (2.6)	0.11	-

^1^ Three students did not provide their date of birth, one student refused to be weighed and measured, for one student fat and muscle percentage were not recorded, and visceral fat was not displayed on the body composition monitor for 40 students. ^2^ H = Hairdresser, A = Admin, S = Sports, N = Nursing.

**Table 2 ijerph-18-03031-t002:** Significant Bonferroni corrected pair-wise comparisons and corresponding effect sizes.

Difference per Activity Type	Effect Size	95% CI ^1^	*p*-Value
***Sedentary time***			
Hairdresser < Admin	−1.59	−2.25–−0.93	<0.001
Hairdresser < Nursing	−1.19	−1.88–−0.50	0.004
Sports < Admin	−1.43	−2.04–−0.82	<0.001
Sports < Nursing	−1.03	−1.77–−0.29	0.036
***Standing time***			
Hairdresser > Admin	1.87	1.20–2.54	<0.001
Hairdresser > Sports	1.11	0.46–1.75	0.004
Hairdresser > Nursing	0.83	0.15–1.50	0.09
Sports > Admin	0.76	0.18–1.34	0.06
Nursing > Admin	1.04	0.32–1.76	0.03
***Stepping time***			
Sports > Hairdresser	1.51	0.85–2.17	<0.001
Sports > Admin	1.41	0.80–2.01	<0.001
***Time in bed***			
Nursing < Hairdresser	−1.26	−1.95–−0.57	0.002
Nursing < Admin	−1.08	−1.81–−0.37	0.02
Nursing < Sports	−1.36	−2.11–−0.61	0.002

^1^ 95% confidence interval.

**Table 3 ijerph-18-03031-t003:** Zero-order correlations between the PA/SB variables and the cognitive outcome variables.

	Inhibition (*n* = 82)	Shifting (*n* = 82)	Updating (*n* = 70)
Sitting	0.032	−0.055	0.144
Standing	0.018	0.138	−0.147
Stepping	−0.060	0.122	0.034

Note: These are Pearson’s r correlation coefficients. The analyses were also performed using Kendall’s tau b and Spearman’s rho, leading to the same conclusion: no significant associations.

## Data Availability

The datasets analyzed during the current study are available from the corresponding author upon reasonable request.

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
