# Peer review of "Differences in Habitual Physical Activity Behavior between Students from Different Vocational Education Tracks and the Association with Cognitive Performance"

_ijerph, 2021, doi:10.3390/ijerph18063031_

Round 1
Reviewer 1 Report
Dear authors,
This study aims to investigate a novel and interesting area. The introduction section provides a lot of information and highlights the lack of studies examining the association between PA and EF in VET students. Also, an important aspect of this study is the inclusion of accelerometry as a more objective insight in PAB in comparison with previous studies.
All materials section is very detailed, and the discussion is coherent. However, I would like to provide some minor concerns:
Abstract section
L20: I would suggest adding in the abstract the meaning of PA and SB in its first time appearing.
Introduction section
L47: The authors mentioned the purpose of this study. However, the information will be repeated in the L113 of the introduction section. Repeated information should be avoided in order to become the introduction easier for the readers. I suggest to state the aims of the study only in the end of the introduction (L113).
L60: "Hence, it is likely to expect distinct differences in PAB of students studying in different educational tracks." In this sentence, the expression "students studying" is redundant. I suggest "students from different education tracks".
Materials section
L293-295: More information in regard to body composition assessment should be given. Although body composition variables not the main variables in the current manuscript, some information regarding to the equipment used and the conditions of the assessment would be important.
Discussion section
L:440 Please insert reference for these statements: “besides differences attributable to curricula, PAB patterns could also reflect preferences of students, independent of curricular PAB. For example, Sports track student may be more involved in sports and exercise and thus show higher amounts of MVPA”. It is not clear if the statements abovementioned are from the reference above.
L488: I would recommend removing the commas. The sentence sounds better without commas. However, it is only a suggestion.
L491: "Although the literature suggests relations between PAB variables EF (...)" - Did the authors want to say between PAB and EF variables?
Best regards.
Author Response
Dear reviewer,
We would like to thank you for your positive comments, your critical review of our manuscript, and for making valuable suggestions. Below we provide point-by-point answers to the comments. Please note that the text that is presented in Italics is as it is presented in the revised manuscript. In the manuscript itself, changes are marked in yellow.
Kind regards on behalf of all co-authors,
Dr. Rianne Golsteijn

Reviewer 2 Report
This study set out to measure sedentary behavior and physical activity in Dutch vocational students, and reported overall difference in these behaviors according to vocational track. Additionally they measured cognitive performance via on-line tests of working memory and executive function, and showed no association of these cognitive tests to sedentary or physical activity.
This appears to be a novel study, since few studies have assessed physical activity according to vocational track, and none have linked objective daily activity to cognitive performance in such a population. Main comments: While the activity measurement appears to be well-designed, the assessment of cognitive performance needs to be carefully controlled for environmental factors such as consumption of caffeine or prior sleep or exercise before testing. Due to the nature of on-line testing this can be difficult to check in subjects, but this needs to be addressed as a limitation of the study It may be interesting to also assess cognitive function according to vocational track, or BMI. detailed comments: page 2 line 68 "percentages of 67 VET students that are sufficiently physically active" what is the threshold for sufficiently active? page 2, line 72: I'm not sure if total screen time is a good measure of leisure sedentary time, since a lot of work is with screens. Was work-related screen time excluded? page 3 line 139 describes what encompasses Sports as a vocational track page 5 line 233 why were these cognitive tests used? Were there previous correlations to PA? page 7 line 289 weight and health were checked, but not clear how this data was taken into account in the cognitive performances analyses. page 8, line 335 Were there differences in demographics between any of the groups (age, gender etc) which might have impacted PA? page 10 line 388 interesting data, but error bars should be depicted in figure page 11 line 413 Table title: This table doesn't have any data on age, please correct -Spelling and grammar errors were noted in the paper (for example page 13, line 526, Strength is mis-spelled). Content needs to be thoroughly proof-read.
Author Response

(The authors gave the same response as above.)

Reviewer 3 Report
Thank you for giving me to review your manuscript. This manuscript is interesting and scientifically meaningful for considering the relationship between physical activity behavior and vocational education track among students. Regarding the contents, I have several suggestions.
- In the abstract, what is PA and SB? Abbreviation should be clarified at the first appearance in the text.
- In the sample section of the method, there are no descriptions regarding sample calculation. The authors should descript why they decided the sample size.
- In the method section, the authors should include the inclusion and exclusion criteria.
- Figure 1 contains some abbreviations, and it can be difficult for potential readers to understand. The authors should add explanations as a footnote.
- The discussion part should start with a summary of the results and outstanding points of this research.
Author Response

(The authors gave the same response as above.)
